# Differentially Private Set Representations

**Sarvar Patel**[*]
Google
sarvar@google.com

**Giuseppe Persiano**[*]
Università di Salerno
giuper@gmail.com

**Joon Young Seo**[*]
Google
jyseo@google.com

**Kevin Yeo**[*]
Columbia University, Google
kwlyeo@google.com

## Abstract

We study the problem of differentially private (DP) mechanisms for representing sets of size $k$ from a large universe. Our first construction creates $(\epsilon, \delta)$-DP representations with error probability of $1/(e^\epsilon + 1)$ using space at most $1.05k\epsilon \cdot \log(e)$ bits where the time to construct a representation is $O(k \log(1/\delta))$ while decoding time is $O(\log(1/\delta))$. We also present a second algorithm for pure $\epsilon$-DP representations with the same error using space at most $k\epsilon \cdot \log(e)$ bits, but requiring large decoding times. Our algorithms match our lower bounds on privacy-utility trade-offs (including constants but ignoring $\delta$ factors) and we also present a new space lower bound matching our constructions up to small constant factors. To obtain our results, we design a new approach embedding sets into random linear systems deviating from most prior approaches that inject noise into non-private solutions.

## 1 Introduction

Consider the problem of releasing a set $S$ of elements from a potentially very large universe $U$ in a differentially privately manner. The goal is to construct a differentially private representation of $S$, denoted by $\hat{S}$. The representation $\hat{S}$ can be used to try and determine whether an element $u \in U$ belongs to the original input set $S$. $\hat{S}$ may err in two ways. For any $u \in S$, $\hat{S}$ may report a false negative stating that $u$ is not in $S$. Also, for $u \notin S$, $\hat{S}$ may report a false negative claiming $u$ appears in $S$. Ideally, we should minimize the error probability for maximal utility while obtaining strong privacy for $S$. This problem is useful for applications where users wish to privately disclose information such as sets of bookmarked websites, visited IP addresses, installed mobile apps, etc. One particularly important application is training machine learning models using the above examples as feature vectors while maintaining user privacy. As some concrete examples, good solutions to our problem could enable privately training models for web traffic forecasting using user's visited webpages [26], app install predictions with user's installed apps sets [5] and detecting shared IP addresses from user's visited IP addresses [17].

A naive approach is to interpret the universe $U$ as a bit vector where each element corresponds to a unique entry of the vector $\mathbf{v} \in \{0, 1\}^{|U|}$. Encoding $S \subseteq U$ works by setting the corresponding coordinates of $S$ to 1 and the rest to 0. Then, we can apply randomized response [33] to each entry of $\mathbf{v}$. The noisy vector $\mathbf{v}$ is then released as the encoding of $S$. Accessing an element proceeds by reading the value at the corresponding coordinate of the noisy vector $\mathbf{v}$. With this approach, the

---

[1] [*]The authors are listed in alphabetical order.

encoding size scales linearly with $|U|$. In most practical applications, the universe $U$ is very large while the input set $S$ is quite small. For example, one can consider $S$ to be set of visited websites. The universe $U$ will be the set of all websites that will be impractically large to store while $S$ will be a much smaller set. Our goal is to design a construction whose size and encoding time depends only on the input set $S$ size while maintaining small error matching randomized response.

To tackle this problem, prior works started non-private solutions for efficiently representing sets such as Bloom filters [8]. First, the input set is encoded using the non-private solution. Afterwards, each entry of the resulting representation is perturbed by injecting noise according to some distribution. This approach was studied in [1] where they showed the encoding was private. However, their work lacked any analysis on the error probability beyond empirical evaluation. To our knowledge, no other work has studied DP representations of sets.

Another related line of work considers DP mechanisms for releasing sparse histograms. In this problem, each element of the input set $S$ is also associated with some value. The goal of a query is to decode the value associated with the queried element (if it exists in the input set $S$). Sparse histograms may also be interpreted as a sparse vector problem where the input vector has at most $k$ non-zero entries. Unlike the private set problem, sparse histograms and vectors have been heavily studied. The majority of these works also take the same approach building on top of (potentially) non-private solution and injecting noise to the resulting representation. For example, several works study count sketch [24, 28, 34, 35] and count min-sketch [27, 21] where each entry of the resulting sketch is perturbed by some DP mechanism. To our knowledge, the only work that slightly deviates from this approach is [2], but they still inject noise using randomized response on bit-level representations (and the Laplacian mechanism in certain settings).

One could attempt to use sparse histograms (or vectors) to represents sets. We can associate each element in the input set $S$ with the value $1$. To decode, we could round the output decoding of the underlying sparse histogram algorithm to either $0$ or $1$. Unfortunately, the error probability guarantees are unclear directly using prior analysis. For example, many prior works show that the per-entry error is at most $O(1/\epsilon)$. That is, the true value and noise value differ by at most $O(1/\epsilon)$. However, it is unclear how this can be directly translated into error probability. In particular, the exact constants of the per-entry error would need to be known to derive a probability bound of whether the decoded output is closer to $0$ or $1$. As an example, the error probabilities would differ greatly if the per-entry error was at most $1/\epsilon$ as opposed to $100/\epsilon$ when using rounding. In our work, we present constructions using a completely different approach to avoid this technical obstacle. Our solutions obtain better per-entry error (both theoretically and empirically) than prior sparse histograms.

## 1.1 Our results

Our main contributions are efficient constructions for differentially private representations of sets that achieve optimal privacy-utility trade-offs and optimal space usage. In particular, our constructions exactly match the utility achieved by randomized response (even up to constants). Our work deviates from prior approaches that aim to construct some representation and perturb using noise. Instead, we embed the input set into a random linear system. Most elements in $S$ are guaranteed to satisfy their corresponding linear constraint in the linear system. In contrast, all elements outside of the element set $S$ will be unlikely to satisfy the relevant constraint except with small probability (that is a controllable parameter in our algorithms). Our constructions are inspired by retrieval data structures based on linear systems such as [29, 15, 16, 7]. In particular, one can view our work as generalizations of these techniques for differential privacy. We only consider error probabilities $\alpha < 1/2$. When error is $\alpha \geq 1/2$, the task is trivial. One can encode a random hash function (independent of the set size) that is perfectly private with $\epsilon = 0$ and $\delta = 0$ (see Appendix F). We present two constructions: one for each of approximate and pure differential privacy.

**Theorem 1.1** (Approximate-DP). *Let $S \subseteq U$ be a set of size $k$ from a universe of size $n$. For any $\epsilon > 0$ and $\delta > 0$, there exists an $(\epsilon, \delta)$-DP algorithm for representing $S$ with error probability $\alpha = 1/(e^\epsilon + 1)$ and space of $1.05k\epsilon \cdot \log(e)$ bits with three hash functions. The encoding time is $O(k \log(1/\delta))$ and the decoding time is $O(\log(1/\delta))$.*

**Theorem 1.2** (Pure-DP). *Let $S \subseteq U$ be a set of size $k$ from a universe of size $n$. For any $\epsilon > 0$, there exists an $\epsilon$-DP algorithm for representing $S$ with error probability $\alpha = 1/(e^\epsilon + 1)$ and space of $k\epsilon \cdot \log(e)$ bits with one hash function. The encoding time is $O(k \log^2 k)$ and the decoding time is $O(k)$.*

We can compare the error probabilities achieved by our DP set mechanisms compared to prior works. For private histograms, per-entry expected error is $\Omega(1/\epsilon)$ as shown in [23]. In contrast, our constructions err with probability $1/(e^\epsilon + 1)$. Note, we can convert this into the expected per-entry error as $1/(e^\epsilon + 1)$. So, we obtain exponentially smaller per-entry error of $1/(e^\epsilon + 1)$, which is impossible for private histograms. We also perform experimental evaluation in Section 5 to corroborate our error being exponentially smaller compared to private histograms.

**Lower bounds.** We show that our constructions achieve optimality in two important dimensions: trade-offs between privacy and utility as well as privacy and space. First, we present a lower bound on the best possible trade-off between privacy and utility (that is, error probability). Our pure-DP solution matches this lower bound exactly including constants. Similarly, our approximate-DP algorithm matches the lower bound (including constants) if we ignore the $\delta$ factor. We also present a lower bound showing the best possible trade-off between privacy and space (encoding size).

**Theorem 1.3** (Utility-privacy trade-off). *Let $S \subset U$ be a set of size $k$. For any $\epsilon \geq 0$ and $0 \leq \delta \leq 1$, any $(\epsilon, \delta)$-DP algorithm for representing $S$ must have error probability $\alpha \geq (1 - \delta)/(e^\epsilon + 1)$.*

**Theorem 1.4** (Space-privacy trade-off). *Let $S \subset U$ be a set of size $k$. For any $\epsilon \geq 0$ and $0 \leq \delta \leq 1$, any $(\epsilon, \delta)$-DP algorithm for representing $S$ with error probability $0 < \alpha < 1/2$, the encoding bit size must be $\min(\Omega((1 + \delta/e^\epsilon) \cdot k \cdot \log((1/\alpha) - 1)), \log \binom{n}{k})$.*

We can consider the space lower bound restricted to algorithms that obtain the optimal privacy-utility trade-off as well. Therefore, we can set $\alpha = (1 - \delta)/(e^\epsilon + 1)$ into the above lower bound. Assuming standard values of very small $\delta$, we can see that the lower bound becomes $\Omega(k \log(1/\alpha)) = \Omega(k \cdot \epsilon)$. Note that our constructions use space of $1.05k\epsilon \cdot \log(e)$ and $k\epsilon \cdot \log(e)$ bits respectively with error probability $\alpha = 1/(e^\epsilon + 1)$. In other words, the space usage asymptotically matches our lower bound for all reasonable parameter choices of $\delta$. In our proof, we work out the exact constants and show that the constant in the lower bound approaches $\log(e)$ for larger values of $\epsilon$. In fact, we show that both our constructions exactly match the lower bound up to a very small constant of at most $4$ that only occurs when $\epsilon = 0$. Furthermore, we note our lower bounds also apply to probabilistic filters (such as Bloom filters) that could also emit false negative errors.

## 1.2 Related work

**Private filters.** Bloom filter [8] is a space efficient, probabilistic data structure that can be used to test whether an element is a member of a set. [1] show that flipping each bit of a Bloom filter with probability $1/(1 + e^{\epsilon/t})$ is $\epsilon$-DP where $t$ is the number of hash functions. However, their work only experimentally evaluates the utility without any provable guarantees. Additionally, we note that prior works have attempted to analyze the privacy properties of filter data structures without modification. For example, this has been studied for Bloom filters [6], counting Bloom filters [31] as well as groups of multiple filter data structures [30]. In general, the conclusion is that filter data structures without modification fail to obtain reasonable privacy guarantees. Finally, we note Bloom filters have also been used in other differential privacy contexts such as RAPPOR [20] where the goal is to aggregate discrete value responses from clients with local DP.

**Private sparse histograms and vectors.** A histogram is a frequency vector where each coordinate may take on real values. It is known that histograms can be made differentially private by adding Laplacian noise to each coordinate [18]. The expected error of each entry is $O(1/\epsilon)$ where $\epsilon$ is the privacy parameter, and it was shown that this privacy-utility trade-off is essentially optimal [23, 4].

Several works have considered the setting where the histogram is sparse and at most $k$ out of $d$ coordinates are non-zero. The goal is to release a representation of the histogram whose size does not depend on $d$. Compared to the Laplacian mechanism, earlier works either suffered from significantly worse privacy-utility trade-offs [25, 13] or incurred very slow access time [3]. More recently, Aumuller *et al.* [2] proposed an ALP mechanism that achieves expected error of $O(1/\epsilon)$ (matching the lower bound asymptotically) with access time of $O(1/\delta)$. The space usage is also very efficient, obtaining $O(k \log(d + u))$ bits where $u$ is the upper bound on the value of the entries.

Another line of work considers private versions of count sketch, introduced in [12], which can be viewed as a generalization of the Bloom filter. Each element in the set has an associated frequency, and the goal is to estimate the frequency of any element in the universe. Viewing the set as a sparse vector of frequencies, the basic idea of the count sketch is to transform the sparse vector $\mathbf{x} \in \mathbb{N}^d$ to a lower dimensional vector via an affine transformation $\mathbf{Ax} \in \mathbb{N}^D$, where $\mathbf{A}$ is a random matrix

from a specific distribution. From $\mathbf{A}\mathbf{x}$, each coordinate $\mathbf{x}_i$ can be estimated with error that depends on $D$ and the norm of $\mathbf{x}$. Several works [24, 28, 34, 35] analyze the privacy-utility trade-off of the private count sketch with different noise distributions in the context of estimating the frequencies of the elements. Due to the linearity of count sketch, these works also studied the problem in the local model where the histogram is distributed amongst multiple parties. These works consider a more general problem setting than ours. As discussed earlier, it is not immediately obvious how the error guarantees of private count sketch will translate to our problem setting.

## 2 Preliminaries

**Notation.** Throughout our paper, we will use $\ln x$ to denote natural (base-$e$) logarithms and use $\log x$ to denote base-2 logarithms. We denote $[x]$ as the set $\{1, \ldots, x\}$ for any integer $x \geq 0$. We denote all vectors in lower case boldface $\mathbf{x}$ and matrices in capital case boldface $\mathbf{M}$. We denote $\mathbf{x}[i]$ as the $i$-th entry of $\mathbf{x}$. Similarly, we denote $\mathbf{M}[i][j]$ as the $j$-th entry of the $i$-th row vector of $\mathbf{M}$, $\mathbf{M}[i]$ as the $i$th row vector, and $\mathbf{M}[:][j]$ as the $j$th column vector. We denote $\mathbf{x}[a:b]$ as the subvector of $\mathbf{x}$ in range $[a, b]$. We use $\mathbf{x}^{\intercal}$ as the transpose of $\mathbf{x}$. We use $\mathbb{F}^n$ to denote the set of all column vectors of length $n$ over a field $\mathbb{F}$ and $\mathbb{F}^{n \times m}$ to denote the set of all $n$ by $m$ matrices over a field $\mathbb{F}$. We use the notation $\mathbf{1}_{x \in S}$ such that $\mathbf{1}_{x \in S} = 1$ if and only if $x \in S$ and $\mathbf{1}_{x \in S} = 0$ otherwise when $x \notin S$. Finally, given a countable set $S$, we will use $S_i$ to denote the $i$-th element in $S$ (in arbitrary order). The subscript is simply used as a label to distinguish the elements.

**Differential privacy.** The notion of differential privacy (DP) was introduced by [18]. DP algorithms guarantee that small changes to the input will not drastically change the output probability distribution. In other words, two similar (or nearby) inputs will result in very similar output distributions.

Throughout our work, our inputs will be sets $S$ from a universe $U$, $S \subseteq U$. We measure the distance between two sets $S$ and $S'$ as the symmetric set difference that we denote as $S \Delta S' = |S \setminus S'| + |S' \setminus S|$. This is the number of elements that appear in exactly one of $S$ and $S'$. One can interpret the symmetric set difference as the minimum number of elements that need to be added or removed to obtain $S'$ from $S$ (or vice versa). We say that two input sets are neighboring when their symmetric set difference is one, that is, $S \Delta S' = 1$. For convenience, we will denote the distance between two sets $S$ and $S'$ as $|S - S'| = S \Delta S'$ to conform with standard differential privacy notation.

We note that one can also interpret the above using $\ell_1$ distances between vectors. For every entry $u \in U$, we can denote with a unique integer from the set $[|U|]$. Suppose, we use a function $z : U \to [|U|]$ as this mapping. For any set $S \subseteq U$, we map $S$ to the vector $\mathbf{x}_S \in \{0, 1\}^{|U|}$ such that $\mathbf{x}_S[i] = 1$ if and only if there exists $u \in S$ such that $z(u) = i$. With this interpretation, we note that the symmetric set difference between two sets $S$ and $S'$, $S \Delta S'$, is identical to the $\ell_1$ distance between the corresponding vectors defined as $|\mathbf{x}_S - \mathbf{x}_{S'}|_1 = \sum_{i \in [|U|]} |\mathbf{x}_S[i] - \mathbf{x}_{S'}[i]|$.

We present the definition of differential privacy following standard definitions [19].

**Definition 2.1.** A randomized algorithm $\mathcal{M}$ with domain D is $(\epsilon, \delta)$-differentially private if, for all $R \subseteq \mathsf{Range}(\mathcal{M})$ and for all $x, y \in \mathsf{D}$ such that $|x - y| = 1$, then

$$\Pr[\mathcal{M}(x) \in R] \leq e^{\epsilon} \cdot \Pr[\mathcal{M}(y) \in R] + \delta$$

over the randomness of the algorithm $\mathcal{M}$.

**Differentially private set representations.** We focus on differentially private algorithms for releasing sets $S$ of size at most $\hat{k}$, that is, $S \subseteq U$ such that $|S| \leq \hat{k}$ for some input parameter $\hat{k}$. We will focus on the case where the universe $U$ is substantially larger than the input set $S$.

**Definition 2.2.** An algorithm $\Pi = (\Pi.\mathsf{Encode}, \Pi.\mathsf{Decode})$ for representing sets consists of:

- $\hat{S} \leftarrow \mathsf{Encode}(S)$: The (randomized) encoding takes set $S \subseteq U$ and returns encoding $\hat{S}$.

- $b \leftarrow \Pi.\mathsf{Decode}(\hat{S}, u)$: The decoding takes encoding $\hat{S}$ and element $u \in U$ and outputs $b \in \{0, 1\}$.

The construction (encoding) time is the running time of $\Pi.\mathsf{Encode}$ and the access (decoding) time is the running time of $\Pi.\mathsf{Decode}$. The space is the size of encoding $\hat{S}$.

In other words, an algorithm for releasing sets creates an encoding $\hat{S}$ of a set $S \subseteq U$. Furthermore, the algorithm enables checking whether any element $u \in U$, appears in $S$ using the encoding $\hat{S}$.

Next, we define the utility of the differentially private set problem through its error probability. An error occurs when the decoding algorithm for a query $q \in U$ returns an answer that is inconsistent with the original input set $S$.

**Definition 2.3.** An algorithm $\Pi = (\Pi.\mathsf{Encode}, \Pi.\mathsf{Decode})$ for representing sets has error probability at most $\alpha$ if, for any input set $S \subseteq U$ and any set of queries $Q \subseteq U$,

$$\Pr[\forall q \in Q, \mathbf{1}_{q \in S} \neq \Pi.\mathsf{Decode}(\hat{S}, q)] \leq \alpha^{|Q|}$$

where $\hat{S} \leftarrow \Pi.\mathsf{Encode}(S)$ and the probability is over the randomness of $\Pi.\mathsf{Encode}$.

For any set of queries $Q$, the probability that all $|Q|$ queries are incorrect is at most $p^{|Q|}$. This is a stronger definition than prior works that consider $|Q| = 1$ because it also ensures independence of incorrect answers. For example, consider any two queries $q_1 \neq q_2 \in U$. Each of them must be incorrect with probability at most $\alpha$ by setting $Q = \{q_1\}$ or $Q = \{q_2\}$. Furthermore, they must be independent since the probability that they are both incorrect is at most $\alpha^2$ by setting $Q = \{q_1, q_2\}$. This independence argument may be extended to arbitrary query set with more than two queries.

We can also interpret this definition as per-entry expected error used in private histograms that bounds the absolute value between the true and decoded value. Our definition may be viewed as privately encoding an $|U|$-length binary vector such that $\mathbf{E}[|\mathbf{1}_{q \in S} - \Pi.\mathsf{Decode}(\hat{S}, q)|] \leq \alpha$ for any element $q \in U$ and encoding $\hat{S} \leftarrow \Pi.\mathsf{Encode}(S)$. In other words, the expected per-entry error is at most $\alpha$.

# 3 Differentially private sets

In this section, we present our main two constructions for differentially private sets. Before we present our constructions, we present a framework for building these algorithms using linear systems that satisfy certain properties. In particular, our work is inspired and generalizes prior retrieval data structures based on linear systems such as [29, 15, 16, 7]. Afterwards, we instantiate the linear systems in two different ways to obtain our constructions (although, one could use other linear systems as we will provide some examples later).

## 3.1 Framework from linear systems

We present a general framework based on linear systems for building DP set mechanisms. We consider linear systems over a finite field $\mathbb{F}$ with two functions: Row and Solve.

Recall that our problem is to release differentially private representation of $S \subseteq U$ such that $|S| \leq \hat{k}$, where $\hat{k}$ is the input to the algorithm. We assume $\mathsf{Row} : U \to \mathbb{F}^{1 \times m}$ is a hash function mapping universe elements to row vectors of length $m$. Here, the parameter $m$ is a function of $\hat{k}$ and does not depend on the size of the input set $S$. Given a set $S = \{s_1, \ldots, s_k\} \subseteq U$ of $k \leq \hat{k}$ elements, one can view Row as hashing $S$ to a $k \times m$ matrix:

$$\mathbf{M} = \begin{bmatrix} \mathsf{Row}(s_1) \\ \ldots \\ \mathsf{Row}(s_k) \end{bmatrix}.$$

The algorithm Solve takes a matrix $\mathbf{M} \in \mathbb{F}^{k \times m}$ and solution vector $\mathbf{b} \in \mathbb{F}^k$ to compute the solution $\mathbf{x} \in \mathbb{F}^m$ satisfying $\mathbf{M}\mathbf{x} = \mathbf{b}$. In particular, Solve will make the assumption that $\mathbf{M}$ is the generated output of Row for some set $S \subseteq U$ of size $k$. For our chosen linear systems, Solve will be faster than the naive application of Gaussian elimination. We also make some additional assumptions about Solve. First, we will exclusively focus on the case where the matrix has more columns than rows, $n \geq k$. Secondly, if the input matrix $\mathbf{M}$ does not have full rank, then Solve will return $\bot$. Lastly, all free variables will be set to uniformly random elements from $\mathbb{F}$.

We note that Row will generate rows in some structured way depending on the chosen linear system to ensure Solve successfully outputs a solution with high probability assuming the number of columns $m$ is sufficiently larger compared to the number of rows $k$. In our work, we focus on two constructions:

random band [15] and Vandermonde matrices. Although, our framework is compatible with any linear system.

We will also use a hash function $h : U \to \mathbb{F}$ that maps each element in the universe $U$ to elements in $\mathbb{F}$. We will use $h$ to generate the solution vector $\mathbf{b}$ in the above linear system. For some noised input set $S = \{s_1, \ldots, s_k\} \subseteq U$, the solution vector will be $\mathbf{b} = [h(s_1), \ldots, h(s_k)]^{\mathsf{T}}$.

In our work, we will assume that all hash functions are fully random following prior works including [28, 34, 35]. In practical implementations, we use cryptographic hash functions to replace this assumption as done in the past [14, 34]. Specifically, we will assume that $h$ and Row are fully random when necessary (for one of our constructions, Row will be deterministic).

**Encoding.** Suppose we are given an input set $S = \{s_1, \ldots, s_k\} \subseteq U$ of size $|S| = k$. First, we generate random hash function $h$ and (possibly random) row function Row. Next, we will randomly sample a subset $S' \subseteq S$ such that each element of $S$ will appear in $S$ except with some *exclusion probability* $p$ (that we pick later during analysis). For convenience, denote $S' = \{s'_1, \ldots, s'_{k'}\}$ where $k' = |S'|$. Encoding works by constructing a matrix $\mathbf{M}$ using Row and noisy input set $S'$ as $\mathbf{M} = [\mathsf{Row}(s'_1), \ldots, \mathsf{Row}(s'_{k'})]^{\mathsf{T}}$. Next, a solution vector $\mathbf{b}$ is created by hashing each of the elements in $S'$ using the hash function $h$. So, $\mathbf{b} = [h(s'_1), \ldots, h(s'_{k'})]^{\mathsf{T}}$. Finally, we compute encoding $\mathbf{x}$ using Solve for the following linear system:

$$
\mathbf{M}\mathbf{x} = \begin{bmatrix} \mathsf{Row}(s'_1) \\ \ldots \\ \mathsf{Row}(s'_{k'}) \end{bmatrix} \cdot \mathbf{x} = \begin{bmatrix} h(s'_1) \\ \ldots \\ h(s'_{k'}) \end{bmatrix}.
$$

The final encoding will be $\hat{S} = (\mathbf{x}, h, \mathsf{Row})$. See Algorithm 1 for formal pseudocode.

---

**Algorithm 1** DPSet.Encode algorithm

**Require:** $S, p, m$: input, exclusion probability $p$, output length $m$
**Ensure:** $\hat{S}$ : DP encoding of $S$
  Generate random hash function $h : U \to \mathbb{F}$.
  Generate (random) Row $: U \to \mathbb{F}^{1 \times m}$.
  $S' \leftarrow \{\}$
  **for** $s \in S$ **do**
    Add $s$ to $S'$ with probability $1 - p$
  **end for**
  $\mathbf{M} \leftarrow |S'| \times m$ matrix $\mathbb{F}^{|S'| \times m}$
  $\mathbf{b} \leftarrow$ length $|S'|$ column vector.
  **for** $i \in [|S'|]$ **do**
    $\mathbf{M}[i] \leftarrow \mathsf{Row}(S'[i])$
    $\mathbf{b}[i] \leftarrow h(S'[i])$
  **end for**
  $\mathbf{x} \leftarrow \mathsf{Solve}(\mathbf{M}, \mathbf{b})$
  **if** $bx \neq \perp$ **then**
    **return** $\hat{S} \leftarrow (\mathbf{x}, \mathsf{Row}, h, \perp)$
  **else**
    **return** $(\perp, \perp, \perp, S)$
  **end if**

---

**Algorithm 2** DPSet.Decode algorithm

**Require:** $\hat{S} = (\mathbf{x}, \mathsf{Row}, h, S), u$
**Ensure:** returns $b \in \{0, 1\}$
  **if** $S \neq \perp$ **then**
    **return** $u \in S$
  **end if**
  $y \leftarrow \mathsf{Row}(u) \cdot \mathbf{x}$
  **return** $\mathbf{1}_{y = h(u)}$

---

We can view the above as using the linear system to embed linear constraints that are satisfied by elements of the noisy input set $S'$. For every $s' \in S'$, we know that $\mathsf{Row}(s') \cdot \mathbf{x} = h(s')$ assuming Solve succeeded. In contrast, fix any $u \notin S'$. Then, we can see that $\Pr[h(u) = \mathsf{Row}(u) \cdot \mathbf{x}] = |\mathbb{F}|^{-1}$ since $h$ is a random hash function. So, elements outside of the set $S'$ are unlikely to be satisfy their corresponding linear constraint. We control this probability by picking the field size $|\mathbb{F}|$ accordingly.

We will capture the event of Solve failing using $\delta$. For our pure-DP construction, we guarantee that $\delta = 0$ and Solve never fails. For our approximate-DP algorithm, we rely on the fact that Solve succeeds with high probability assuming Row is randomly generated in a correct manner. If Solve fails, we assume that encode simply returns the input set $S$.

**Decoding.** Suppose we are given an encoding $\hat{S} = (\mathbf{x}, h, \mathsf{Row})$ and an element $u \in U$. Decoding checks whether an element's corresponding linear system is satisfied by computing $\mathsf{Row}(u) \cdot \mathbf{x}$ and comparing with $h(u)$. In other words, the decoding algorithm simply returns $\mathbf{1}_{\mathsf{Row}(u) \cdot \mathbf{x} = h(u)}$. We present the pseudocode in Algorithm 2.

We start by presenting the error probability (utility) with respect to field size $|\mathbb{F}|$ and the exclusion probability $p$ of removing any element. We defer the proofs to Appendix A.

**Theorem 3.1.** *If $|\mathbb{F}| = \alpha^{-1}$ and $p = \alpha/(1 - \alpha)$, then* DPSet.Decode *has error probability $\alpha$.*

For error probability $\alpha$, we pick $|\mathbb{F}| \geq 1/\alpha$ holds where $\mathbb{F}$ is a finite field. We note that there is a finite field of size $q^r$ for any prime $q$ and positive integer $r \geq 1$. For practical purposes, we use the smallest integer $q^r$ larger than $1/\alpha$ that gives us slightly smaller error probability.

Next, we prove privacy of our framework. We defer the full proof to Appendix B.

**Theorem 3.2.** *If* Solve *errs with probability at most $\delta$, then* DPSet *is $(\epsilon, \delta)$-DP with error $(e^\epsilon + 1)^{-1}$.*

Our construction's expected per-entry error of $\alpha = 1/(e^\epsilon + 1)$ is exponentially smaller than achievable by private histograms where $\Omega(1/\epsilon)$ error is required [23].

Next, we analyze the encoding size. In general, these are largely dependent on the underlying linear system. The encoding size depends on the number of variables (columns) $m$ in the linear system. Additionally, it also includes representations of the functions $h$ and $\mathsf{Row}$.

**Theorem 3.3.** DPSet.Encode *outputs encodings of $m$ field elements and encodings of $h$ and* $\mathsf{Row}$.

In Appendix E, we outline a possible optimization to reduce encoding size by picking $m$ closer to the expected size of the sampled set $S'$. This turns out to be a more theoretical as we were unable to observe space improvements empirically for reasonable choices of set size $k$ and error probability $\alpha$.

**Computational time.** For computation, the majority of the work is done by the underlying linear system. In particular, DPSet.Encode requires only $O(k)$ time outside of Solve and $\mathsf{Row}$. Similarly, DPSet.Decode requires an execute of $\mathsf{Row}$ and the computation will depend on the number of non-zero entries in $\mathsf{Row}$. We analyze the computational costs for our instantiations later.

**Larger error of $\alpha > 1/2$.** Our constructions only consider error probabilities $\alpha \leq 1/2$. This is implicit as the smallest field has size at least 2. There are trivial algorithms to obtain mechanisms with $\epsilon = 0$ and $\delta = 0$ for the case of $\alpha \geq 1/2$ using a random hash function (see Appendix F).

## 3.2 Approximate differentially private sets

From Section 3.1, our goal essentially boils down to constructing a linear system where a solution exists and may be efficiently computed with high probability. Furthermore, we want to minimize the number of variables required to ensure small encoding sizes. To this end, we will use the *random band row vector* construction of [15].

The random band construction is parameterized by the row length $m$ and the band length $w$. At a high level, each row consists of a single band of $w$ random field elements. The band's location is chosen uniformly at random. All $m - w$ entries outside of the band will be zero. Formally, the construction uses hash functions $h_1 : U \rightarrow [m - w + 1]$ and $h_2 : U \rightarrow \mathbb{F}^{1 \times w}$. For $u \in U$, $h_1(u)$ denotes the band's starting location and $h_2(u)$ is the $w$ elements in the band. Generating a random $\mathsf{Row}_{\mathsf{band}}$ is equivalent to generating the two random hash functions $h_1$ and $h_2$. $\mathsf{Solve}_{\mathsf{band}}$ works by sorting the rows by starting band location and executing Gaussian elimination. See Algorithms 3 and 4.

---

**Algorithm 3** $\mathsf{Row}_{\mathsf{band}}$ algorithm

**Require:** $u$: element $u \in U$
**Ensure:** $\mathbf{v}$ : random band row vector
  $m \leftarrow$ length of the vector
  $\mathbf{v} \leftarrow 0^{1 \times m}$ (all zero row vector of length $m$)
  $s \leftarrow h_1(u)$
  $\mathbf{v}[s : s + w - 1] \leftarrow h_2(u)$
  **return** $\mathbf{v}$

---

**Algorithm 4** $\mathsf{Solve}_{\mathsf{band}}$ algorithm

**Require:** $\mathbf{M}, \mathbf{b}$: matrix and vector
**Ensure:** $\mathbf{x}$ : solution satisfying $\mathbf{Mx} = \mathbf{b}$
  Sort rows by starting band location.
  Execute Gaussian elimination and set free variables to be random elements in $\mathbb{F}$ to obtain encoding $\mathbf{x}$.
  **return** $\mathbf{x}$

---

At a high level, If $k$ is the number of rows of the matrix, Dietzfelbinger and Walzer [15] showed that if $m = (1 + \beta)k$ and $w = O(\log k)$ for some constant $\beta > 0$, then the matrix generated using the random band row construction has full rank and $\mathsf{Solve_{band}}$ runs in $O(mw)$ time except with probability $O(1/m)$. Bienstock *et al.* [7] extended this result to show that, if $w = O(\log(1/\delta) + \log k)$, then the matrix has full row rank and the linear system can be solved in time $O(mw)$ except with probability $\delta$. $\mathsf{DPSet.Decode}$ takes $O(w)$ time since computing the dot product scales linearly with $w$, the length of the band. We obtain the following using random band row vectors:

**Theorem 3.4.** *For any $\epsilon > 0$, $\delta > 0$, $\beta > 0$, there is an $(\epsilon, \delta)$-DP set mechanism with error $(e^\epsilon + 1)^{-1}$ and encodings consisting of $(1 + \beta)k$ field elements and three hash functions.* $\mathsf{DPSet.Encode}$ *takes $O(kw)$ time and* $\mathsf{DPSet.Decode}$ *takes $O(w)$ time where $w = O(\log(1/\delta) + \log k)$.*

### 3.3 Pure differentially private sets

We consider a pure differentially private construction of the framework in Section 3.1 with $\delta = 0$. In Section 3.1, the failure probability of solving the constructed linear system corresponds to $\delta$ in the DP definition. To obtain a pure DP construction, our goal is to construct a linear system that is solvable with probability 1. So, we want to construct a matrix $\mathbf{M}$ that has full rank with probability 1. To do this, we use the *Vandermonde matrix* construction (where Row is deterministic) that may be solved in $O(k \log^2 k)$ time as shown in [9]. This construction has another advantages over the random band approach beyond obtaining $\delta = 0$. The resulting encodings are smaller with only $k$ field elements whereas the other construction requires $m = (1 + \beta)k$ field elements with $\beta > 0$. In contrast, decoding times are larger here. See Appendix C for full description and proof.

**Theorem 3.5.** *For any $\epsilon > 0$, there exists an $\epsilon$-DP set mechanism with error $(e^\epsilon + 1)^{-1}$.* $\mathsf{DPSet.Encode}$ *takes $O(k \log^2 k)$ time and* $\mathsf{DPSet.Decode}$ *takes $O(k)$ time.*

**Other Constructions.** We present two concrete constructions from specific linear systems, but it is possible to plug in other linear systems. For example, plugging in [22] would result in a pure DP solution with faster encoding times, but larger encoding sizes compared to Theorem 3.5.

## 4 Lower bounds

**Privacy-utility lower bounds.** We start by considering the possibility of improving the error probability (utility) with respect to the desired levels of privacy. Our construction achieved error probability at most $1/(e^\epsilon + 1)$ for any choice of $\epsilon \geq 0$. In other words, for any error $\alpha$, our construction achieves privacy $\epsilon = \log((1 - \alpha)/\alpha)$. We show that this trade-off between $\epsilon$ and error probability is optimal even up to constants (ignoring $\delta$ factors). See Appendix D for the proof.

**Theorem 4.1.** *Consider any $(\epsilon, \delta)$-DP algorithm $\Pi$ for sets of size $k$. Suppose that $\Pi$ has error probability at most $\alpha \leq 1/2$. Then, $\epsilon \geq \ln((1 - \alpha - \delta)/\alpha)$. In other words, for a fixed privacy level $\epsilon \geq 0$ and $\delta \geq 0$, the error probability of $\Pi$ must be $\alpha \geq (1 - \delta)(e^\epsilon + 1)$.*

**Space lower bounds.** Next, we move onto determining the necessary space usage of set representations. There exist space lower bounds for probabilistic membership data structures (such as Bloom filters) that have a false positive probability of $\alpha$ and no false negatives. It is well known that such data structures require $k \cdot \log(1/\alpha)$ bits of space when given an input of size $k$. However, these lower bounds only apply when the false negative rate is 0. See Broder and Mitzenmacher [10] for the prior lower bound. We present a space lower bound for DP mechanisms with non-zero false negatives using a proof through compression that deviates from prior counting arguments (see Appendix D).

**Theorem 4.2.** *Consider any $(\epsilon, \delta)$-DP $\Pi$ for sets of size $k$. If $\Pi$ produces $s$-bit encodings with error probability $0 < \alpha \leq 1/2$, then $\mathbf{E}[s] = \Omega\left((1 + \delta/(e^\epsilon)) \cdot k \cdot \log(1/\alpha)\right)$.*

## 5 Experimental evaluation

**Setup.** We implemented DPSet, ALP [2] and DP Count Sketch [34] in C++ using 800 lines of code. For DPSet, we use the analysis of Bienstock *et al.* [7] to choose appropriate parameters for $\delta \leq 2^{-40}$ with parameter $\beta = 0.05$. To fit ALP and DP Count Sketch to our problem setting, we round the query results of these mechanisms to the nearest 0 or 1. We target privacy parameter $\delta \leq 2^{-40}$ for all

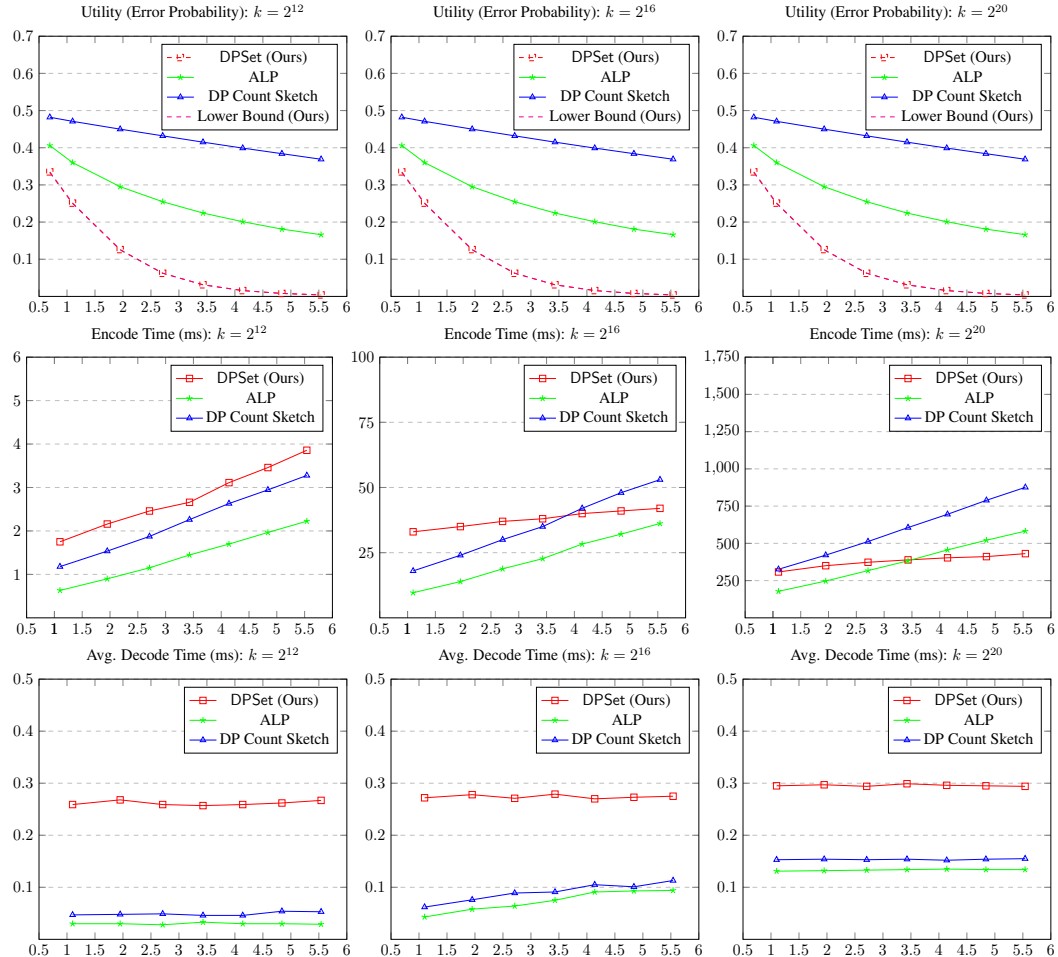

Figure 1: Comparisons of of DPSet, ALP, and DP Count Sketch with $\delta \leq 2^{-40}$. The $x$-axis is privacy parameter $\epsilon$ and the $y$-axis is error probability, encoding time (ms) or decoding time (ms).

three constructions. To fairly compare utility, we chose parameters to ensure that encoding sizes are approximately equal for all three constructions (see Appendix G for more details on encoding sizes).

We consider experiments for input sets of size $k \in \{2^{12}, 2^{16}, 2^{20}\}$. Each trial picks input sets as $k$ uniformly random 128-bit strings from a universe of all $n = 2^{128}$ strings. Although, all three constructions are agnostic to the distribution of the input set. We ran all experiments using a Ubuntu PC with 12 cores, 3.7 GHz Intel Xeon W-2135 and 64 GB of RAM. Our experiments enable AVX2 and AVX-512 instruction sets with SIMD instructions. All reported results use single-thread execution as the average of at least 1,000 trials with standard deviation less than 10% of the average. The entire experimental evaluations (including setup) took approximately 1 hour of compute time.

**Utility.** To measure utility, we query the entire input set of size $k$ as well as a random subset of $k$ elements outside of the set in each trial. We plot our results in Figure 1 along with our lower bound (Theorem 4.1). We see that DPSet has much better utility compared to the prior works. Furthermore, our experiments corroborate our theoretical analysis that error probability exponentially decreases in $\epsilon$ and essentially matches our lower bound of $\alpha \geq (1 - \delta)/(e^{\epsilon} + 1) \geq (1 - 2^{-40})/(e^{\epsilon} + 1)$.

**Efficiency.** We compare the efficiency of encoding input sets and decoding random elements. For larger input set sizes $k$ and bigger $\epsilon$, our constructions have faster encoding times. In contrast, DPSet has slower encoding for smaller $k$ and $\epsilon$. For decoding, DPSet has slower times than both prior works. Nevertheless, decoding times of DPSet remain very fast and are less than $0.3$ milliseconds.

# 6 Conclusions

In this work, we present constructions of DP sets that are essentially optimal in privacy-utility and space trade-offs nearly matching our lower bounds. The error obtained is exponentially smaller (both theoretically and empirically) than possible for private histograms mostly studied in prior works. Additionally, we experimentally show that our constructions are concretely efficient.

**Limitations.** A limitation of our work is that we consider sparse sets (as opposed to the more general sparse histograms). Nevertheless, we believe this specific problem has several important applications with the added benefit of exponentially smaller error. Our constructions assume fully random hash functions (following several prior works) and instantiations are limited to finite field sizes. If we assume pseudorandom hash functions (PRFs), our construction obtains computational DP instead.

# 7 Acknowledgements

The authors would like to thank Rachel Cummings for feedback on earlier versions of this paper.

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

# A   Proof of correctness of DPSet

We prove the correctness of our constructions. First, we present a lemma about the false positive and false negative rates. Afterwards, we use this to get a final error probability.

**Lemma A.1.** DPSet.Decode *has false positive probability of* $|\mathbb{F}|^{-1}$ *and false negative probability of* $p \cdot (1 - |\mathbb{F}|^{-1})$. *Also, the probabilities are independent for each* $q \in U$.

*Proof.* First consider false positives. If $u \notin S$, then $u \notin S'$. We know $\Pr[\text{Row}(u) \cdot \mathbf{x} = h(u)] = |\mathbb{F}|^{-1}$ since $h$ is a fully random hash function. For false negatives, consider any $s \in S$. Note, that $\Pr[s \notin S' \mid s \in S] = (1 - p)$. Since DPSet.Decode only returns 0 if $\mathbf{x} = \bot$, there is no additional error from DPSet.Encode failing. If $s \notin S'$, the decoding will return 1 with probability $|\mathbb{F}|^{-1}$. So, the false negative probability is $s$ not sampled into $S'$ and the linear constraint being unsatisfied that is $p \cdot (1 - |\mathbb{F}|^{-1})$. Finally, these probabilities are independent for every $q \in U$ following from the fact that sampling each element into $S'$ is independent and that $h$ is a fully random hash function.   $\square$

The error probability is the maximum of the false positive and negative probabilities. Suppose we desired a certain error probability $\alpha$, we can pick the field size and exclusion probability as follows.

*Proof of Theorem 3.1.* First, we see that $\alpha = |\mathbb{F}|^{-1}$ for the false positives. Then, we pick $p$ satisfying $\alpha = p \cdot (1 - \alpha)$ for false negatives to see that $p = \alpha/(1 - \alpha)$.   $\square$

# B   Proof of privacy of DPSet

In this section, we present the full proof of Theorem 3.2. In particular, we show that it follows directly from the following theorem by plugging in $\epsilon$ accordingly. In our proof, we require the randomness of $h$ only for correctness and not privacy.

**Theorem B.1.** *Suppose* $|\mathbb{F}| = \alpha^{-1}$, $p = \alpha/(1 - \alpha)$ *and* Solve *fails with probability* $f_{\text{Solve}}$ *for correctly generated* Row. *Then,* DPSet *is* $(\ln(\frac{1-\alpha}{\alpha}), f_{\text{Solve}})$-*DP*.

*Proof.* Let $S_1$ and $S_2$ be the two neighboring sets such that $S_2 = S_1 \cup \{u\}$. Let $\mathbf{Z}_{S_1}$ and $\mathbf{Z}_{S_2}$ be the two random variables denoting the representations output by DPSet.Encode for $S_1$ and $S_2$, respectively. Let $m$ be the number of variables (length of the encoded vector) in Algorithm 1. Let $\mathbf{x} \in \mathbb{F}^m$, Row and $h$ be arbitrary, and let $\mathbf{v} = (\mathbf{x}, \text{Row}, h, \bot)$. We first show that

$$\Pr[\mathbf{Z}_{S_1} = \mathbf{v}] \leq \left(\frac{1 - \alpha}{\alpha}\right) \cdot \Pr[\mathbf{Z}_{S_2} = \mathbf{v}] \tag{1}$$

$$\Pr[\mathbf{Z}_{S_2} = \mathbf{v}] \leq \left(\frac{1 - \alpha}{\alpha}\right) \cdot \Pr[\mathbf{Z}_{S_1} = \mathbf{v}] \tag{2}$$

where the probability is over the random coin tosses performed by DPSet.Encode.

We first prove Equation 1. Let $R_u$ be the event where the element $u$ is removed during DPSet.Encode on $S_2$. Recalling that $\Pr[R_u] = p = \frac{\alpha}{1-\alpha}$ from Algorithm 1, we have

$$\begin{aligned}
&\Pr[\mathbf{Z}_{S_2} = \mathbf{v}]\\
&= \Pr[R_u]\Pr[\mathbf{Z}_{S_2} = \mathbf{v} \mid R_u] + \Pr[\overline{R_u}]\Pr[\mathbf{Z}_{S_2} = \mathbf{v} \mid \overline{R_u}]\\
&= \Pr[R_u]\Pr[\mathbf{Z}_{S_1} = \mathbf{v}] + \Pr[\overline{R_u}]\Pr[\mathbf{Z}_{S_2} = \mathbf{v} \mid \overline{R_u}]\\
&\geq \Pr[R_u]\Pr[\mathbf{Z}_{S_1} = \mathbf{v}] = \frac{\alpha}{1 - \alpha}\Pr[\mathbf{Z}_{S_1} = \mathbf{v}]
\end{aligned}$$

where the second equality follows from the fact that the distribution of $\mathbf{Z}_{S_2}$ is identical to $\mathbf{Z}_{S_1}$ conditioned on the event $R_u$. Rearranging, we get the desired bound.

Next, we prove Equation 2. By again decomposing $\Pr[\mathbf{Z}_{S_2} = \mathbf{v}]$ conditioned on $R_u$, we have

$$\begin{aligned}
&\Pr[\mathbf{Z}_{S_2} = \mathbf{v}]\\
&= \Pr[R_u]\Pr[\mathbf{Z}_{S_2} = \mathbf{v} \mid R_u] + \Pr[\overline{R_u}]\Pr[\mathbf{Z}_{S_2} = \mathbf{v} \mid \overline{R_u}]\\
&= \Pr[R_u]\Pr[\mathbf{Z}_{S_1} = \mathbf{v}] + \Pr[\overline{R_u}]\Pr[\mathbf{Z}_{S_2} = \mathbf{v} \mid \overline{R_u}].
\end{aligned}$$

Before proceeding with the proof, we claim that

$$\Pr[\mathbf{Z}_{S_2} = \mathbf{v} \mid \overline{R_u}] \leq \alpha^{-1} \Pr[\mathbf{Z}_{S_1} = \mathbf{v}]. \tag{3}$$

Then plugging in Equation 3 to the above equation, we get

$$\Pr[R_u] \Pr[\mathbf{Z}_{S_1} = \mathbf{v}] + \Pr[\overline{R_u}] \Pr[\mathbf{Z}_{S_2} = \mathbf{v} \mid \overline{R_u}]$$
$$\leq \Pr[R_u] \Pr[\mathbf{Z}_{S_1} = \mathbf{v}] + \Pr[\overline{R_u}](\alpha^{-1} \Pr[\mathbf{Z}_{S_1} = \mathbf{v}])$$
$$\leq \left(\frac{\alpha}{1-\alpha} + \frac{1-2\alpha}{\alpha - \alpha^2}\right) \Pr[\mathbf{Z}_{S_1} = \mathbf{v}]$$
$$= \frac{1-\alpha}{\alpha} \Pr[\mathbf{Z}_{S_1} = \mathbf{v}].$$

We now prove Equation 3 to complete the proof. Let $\mathbf{S}'_1$ and $\mathbf{S}'_2$ be random variables denoting the set of elements that survived the removal process in DPSet.Encode for input $S_1$ and $S_2$, respectively. Consider an arbitrary subset $S' \subseteq S_1$. We can see that $\Pr[\mathbf{S}'_2 = S' \cup \{u\} \mid \overline{R_u}] = \Pr[\mathbf{S}'_1 = S']$, and so if we show that

$$\Pr[\mathbf{Z}_{S_2} = \mathbf{v} \mid \overline{R_u} \cap (\mathbf{S}'_2 = S' \cup \{u\})] \leq \alpha^{-1} \Pr[\mathbf{Z}_{S_1} = \mathbf{v} \mid \mathbf{S}'_1 = S'] \tag{4}$$

then we can apply the law of total probability to obtain Equation 3.

One way to see why Equation 4 holds is as follows. If the left hand side is 0, then the bound trivially holds, so we may assume that the probability is positive. For any choice of hash functions (that also determine Row), the linear systems generated are uniquely determined by the surviving elements. Thus, conditioned on the event that $\mathbf{S}'_1 = S'$ and $\mathbf{S}'_2 = S' \cup \{u\}$, the generated linear systems are deterministic. Let $L_1$ and $L_2$ be the corresponding linear systems for $S_1$ and $S_2$, respectively. From the condition, it must be that $L_1 \subset L_2$ and $L_2$ has exactly one more equation than $L_1$ that is linearly independent of the other rows. In other words, $L_1$ has exactly one more degree of freedom than $L_2$, which corresponds to an extra free variable. By the construction of Algorithm 1, the free variables are independently and uniformly randomly set to values in $\mathbb{F}$. Thus, the probability that this random free variable is set to the corresponding value in $\mathbf{x}$ (the first component of $\mathbf{v}$) is exactly $1/|\mathbb{F}| = \alpha$. This establishes the inequality (in fact an equality) and completes the proof of Equation 4, from which Equation 2 follows immediately.

From Equation 1 and Equation 2, the proof of the main theorem follows immediately by applying Definition 2.1 and using the failure probability of Solve is at most $\delta = f_{\text{Solve}}$. $\qquad \square$

## C  Pure differentially private subsets

As discussed in Section 3.3, to obtain a pure DP construction, our goal is to construct a linear system that is full rank with probability 1. To achieve this goal, we will use the *Vandermonde matrix* construction. Vandermonde matrix is a $n \times k$ matrix of the form

$$\begin{bmatrix} 1 & u_1 & \dots & u_1^{k-2} & u_1^{k-1} \\ 1 & u_2 & \dots & u_2^{k-2} & u_2^{k-1} \\ & & \vdots & & \\ 1 & u_n & \dots & u_n^{k-2} & u_n^{k-1} \end{bmatrix}$$

where each $u_i \in \mathbb{F}$. If $n \leq k$ then this matrix is always full rank for any set of distinct $u_i$.

Suppose that the universe $U = \mathbb{F}$ for some finite field $\mathbb{F}$. Let $S$ be the input set and let $k = |S|$. Then we can construct the matrix in Algorithm 1 using the Vandermonde matrix construction to obtain a pure differentially private construction. The algorithm for constructing the row vector is presented in Algorithm 5. Using [9], the linear system constructed using the Vandermonde matrix can be solved in $O(k \log^2 k)$ time. We point to the prior work to find the corresponding $\text{Solve}_{\text{Vandermonde}}$. Plugging this into the framework of Section 3.1, we immediately obtain Theorem 3.5.

**Algorithm 5** Row$_{\text{Vandermonde}}$ algorithm

---

**Require:** $u$: element $u \in U = \mathbb{F}$
**Ensure:** $\mathbf{v}$ : Vandermonde matrix row
   $k \leftarrow |S|$, size of the input set
   **return** $\mathbf{v} \leftarrow [1, u, u^2, \ldots, u^{k-1}]$

---

## D   Proof of lower bounds

We start by proving our lower bound of utility stated in Theorem 4.1.

*Proof of Theorem 4.1.* Pick any $\mathbf{x}$ and $\mathbf{y}$ that differ in exactly one entry. Without loss of generality, pick the unique index $i \in [n]$ such that $\mathbf{x}[i] = 0$ and $\mathbf{y}[i] = 1$. Let $\mathbf{Z_x}$ and $\mathbf{Z_y}$ be the random variables denoting the representations output by $\Pi$ for $\mathbf{x}$ and $\mathbf{y}$ respectively. We will consider the probability that $\Pi$ produces a representation such that $\Pi$.Decode outputs $1$ on index $i$ for each of $\mathbf{Z_x}$ and $\mathbf{Z_y}$. Note that $\Pr[\Pi.\mathsf{Decode}(\mathbf{Z}_y, i) = 1] \geq 1 - \alpha$ since $\mathbf{y}[i] = 1$. Similarly, we note that $\Pr[\Pi.\mathsf{Decode}(\mathbf{Z}_x, i) = 1] \leq \alpha$ since $\mathbf{x}[i] = 0$. In other words, we see

$$
\begin{aligned}
1 - \alpha &\leq \Pr[\Pi.\mathsf{Decode}(\mathbf{Z}_y, i) = 1] \\
&\leq e^\epsilon \Pr[\Pi.\mathsf{Decode}(\mathbf{Z}_x, i) = 1] + \delta \\
&\leq e^\epsilon \alpha + \delta.
\end{aligned}
$$

By re-arranging the inequality $1 - \alpha \leq e^\epsilon \alpha + \delta$, we get the desired theorem. $\qquad\square$

To prove our space lower bound stated in Theorem 4.2, we start with proving an intermediate result about the required space for any mechanism (not necessarily differentially private) that has error probability at most $\alpha$. In particular, the existence of such a mechanism enables a very efficient compression algorithm to encode random vectors $\mathbf{x}$ with $k$ non-zero entries.

**Lemma D.1.** *Consider any mechanism $\Pi$ for binary vectors $\mathbf{x} \in \{0,1\}^n$ with at most $k$ non-zero entries, $|\mathbf{x}|_1 \leq k$. If $\Pi$ produces representations using $s$ bits of space in expectation and has error probability at most $0 < \alpha \leq 1/2$, then*

$$
\mathbf{E}[s] \geq (1 - 2\alpha)k \cdot \log\left(\frac{1}{\alpha} - 1\right) - 2\log k - \log\log(en/k).
$$

*Proof.* We will make the assumption that $\Pi$ never produces representations larger than $\log\binom{n}{k}$ bits on any input and any choice of randomness. Note, this is without loss of generality because a trivial representation of binary vectors with $k$ non-zero entries can be done in $\log\binom{n}{k}$ bits with zero error probability. If $\Pi$ violates this assumption, we can modify $\Pi$ to replace any longer encodings with the trivial one that will either maintain or decrease the space usage and error probability.

We consider the following two-party, one-way compression problem between an encoder (Alice) and a decoder (Bob). As input, Alice receives as input a uniformly random vector $\mathbf{x} \in \{0,1\}^n$ conditioned that exactly $k$ entries are non-zero, $|\mathbf{x}|_1 = k$. Alice's job is to encode $\mathbf{x}$ into a single message enabling Bob to correctly decode $\mathbf{x}$. In particular, Alice's goal is to make the message as small as possible. To do this, Alice will utilize the mechanism $\Pi$. At a high level, Alice will use $\Pi$ to construct a representation $\mathbf{x}$ with error probability $\alpha$. Additionally, Alice will send some auxiliary information that will enable Bob to correctly identify the non-zero entries of $\mathbf{x}$ using the answers of $\Pi$. We present the compression algorithm below.

**Alice's Encoding**: Receives $\mathbf{x} \in \{0,1\}^n$ such that $|\mathbf{x}|_1 = k$ and shared randomness $\mathcal{R}$.

1. Construct $\mathbf{Z} \leftarrow \Pi.\mathsf{Encode}(\mathbf{x}; \mathcal{R})$ using randomness $\mathcal{R}$.

2. Set $X = \{i \in [n] \mid \mathbf{x}[i] = 1\}$.

3. Initialize $A \leftarrow \emptyset$ and $B \leftarrow \emptyset$.

4. For all $i \in [n]$:

    (a) If $\Pi.\text{Decode}(\mathbf{Z}, i; \mathcal{R}) = 0$, set $A \leftarrow A \cup \{i\}$.

    (b) Else when $\Pi.\text{Decode}(\mathbf{Z}, i; \mathcal{R}) = 1$, set $B \leftarrow B \cup \{i\}$.

5. Encode $|\mathbf{Z}|$ using $\log \log \binom{n}{k}$ bits.

6. Encode $|X \cap A|$ using $\log k$ bits.

7. Encode $X \cap A$ using $\log \binom{|A|}{|X \cap A|}$ bits.

8. Encode $X \cap B$ using $\log \binom{|B|}{|X \cap B|}$ bits.

9. Compute encoding $E = (|\mathbf{Z}|, \mathbf{Z}, |X \cap A|, X \cap A, X \cap B)$.

**Bob's Decoding**: Receives Alice's encoding and shared randomness $\mathcal{R}$.

1. Decode $|\mathbf{Z}|$ using the first $\log \log \binom{n}{k}$ bits and $\mathbf{Z}$ using the next $|\mathbf{Z}|$ bits.

2. Initialize $A \leftarrow \emptyset$ and $B \leftarrow \emptyset$.

3. For all $i \in [n]$:

    (a) If $\Pi.\text{Decode}(\mathbf{Z}, i; \mathcal{R}) = 0$, set $A \leftarrow A \cup \{i\}$.

    (b) Else when $\Pi.\text{Decode}(\mathbf{Z}, i; \mathcal{R}) = 1$, set $B \leftarrow B \cup \{i\}$.

4. Decode the size of $|X \cap A|$ using the next $\log k$. Additionally, we know that $|X \cap B| = k - |X \cap A|$.

5. Using knowledge of $|A|, |B|, |X \cap A|$ and $|X \cap B|$, decode $X \cap A$ and $X \cap B$.

6. Using knowledge of $A$ and $B$ as well as $X \cap A$ and $X \cap B$, we can decode $X$ and, thus, $\mathbf{x}$.

**Prefix-freeness.** We will later apply Shannon's source coding theorem. However, to do this, it is required that Alice's encoding algorithm is prefix-free. That is, any possible encoding cannot be a strict prefix of any other possible encoding. First, we note that the last components $|X \cap A|$, $X \cap A$ and $X \cap B$ will always be the same length. We use a fixed length to represent $|X \cap A|$. The two sets $X \cap A$ and $X \cap B$ always encode exactly $k$ elements. Therefore, the encoding length will only be different for various sizes of $\mathbf{Z}$. However, we prefix each encoding with the length $|\mathbf{Z}|$. Therefore, any encodings of different lengths (meaning different length $|\mathbf{Z}|$) will be prefix-free. Finally, it is clear that any set of equal length encodings will be prefix-free.

**Encoding length.** The expected length of Alice's encoding is exactly

$$\log \log \binom{n}{k} + \mathbf{E}[s] + \log k + \mathbf{E}\left[\log \binom{|A|}{|X \cap A|} + \log \binom{|B|}{|X \cap B|}\right]$$

as we consider expected space usage $s$ and all of $A, B, X \cap A$ and $X \cap B$ are random variables. Next, we note that the function $f(a, b) \to \binom{a}{b}$ is log-concave for the relevant range $a \geq b \geq 0$ (see [11] for example). Therefore, we can apply Jensen's inequality to obtain

$$\mathbf{E}\left[\log \binom{|A|}{|X \cap A|} + \log \binom{|B|}{|X \cap B|}\right] \leq \log \binom{\mathbf{E}[|A|]}{\mathbf{E}[|X \cap A|]} + \log \binom{\mathbf{E}[|B|]}{\mathbf{E}[|X \cap B|]}.$$

Next, we know that $|X \cap A| + |X \cap B| = k$ and $|A| + |B| = n$. Therefore, we can rewrite

$$\log \binom{\mathbf{E}[|A|]}{\mathbf{E}[|X \cap A|]} + \log \binom{\mathbf{E}[|B|]}{\mathbf{E}[|X \cap B|]} = \log \binom{\mathbf{E}[|A|]}{\mathbf{E}[|X \cap A|]} + \log \binom{n - \mathbf{E}[|A|]}{k - \mathbf{E}[|X \cap A|]}.$$

Next, we see that $\mathbf{E}[|A|] \leq (1 - \alpha)n$ and $\mathbf{E}[|X \cap A|] \leq \alpha k$. Given that $\alpha \leq 1/2$, we immediately see that this is maximized when $\mathbf{E}[|A|] = (1 - \alpha)n$ and $\mathbf{E}[|X \cap A|] = \alpha k$. So, we see that

$$\log \left( \begin{array}{c} \mathbf{E}[|A|] \\ \mathbf{E}[|X \cap A|] \end{array} \right) + \log \left( \begin{array}{c} n - \mathbf{E}[|A|] \\ k - \mathbf{E}[|X \cap A|] \end{array} \right) \le \log \left( \begin{array}{c} (1 - \alpha)n \\ \alpha k \end{array} \right) + \log \left( \begin{array}{c} \alpha n \\ (1 - \alpha)k \end{array} \right).$$

**Complete the lower bound.** To finally complete the proof of the lower bound, we will apply Shannon's source coding theorem [32] that states that the expected length of Alice's prefix-free encoding cannot be smaller than the entropy of Alice's input conditioned on any shared input. First, we see that

$$H(\mathbf{x} \mid \mathcal{R}) = H(\mathbf{x}) = \log \left( \begin{array}{c} n \\ k \end{array} \right).$$

Therefore, we get that Alice's expected encoding length must satisfy

$$\log \log \left( \begin{array}{c} n \\ k \end{array} \right) + \mathbf{E}[s] + \log k + \log \left( \begin{array}{c} (1 - \alpha)n \\ \alpha k \end{array} \right) + \log \left( \begin{array}{c} \alpha n \\ (1 - \alpha)k \end{array} \right) \ge \log \left( \begin{array}{c} n \\ k \end{array} \right).$$

By re-arranging, we see that the following is equivalent by applying linearity of expectation and using Stirling's approximation such that $\binom{n}{k} \le (en/k)^k$.

$$
\begin{aligned}
\mathbf{E}[s] &\ge \log \left( \frac{\binom{n}{k}}{\binom{(1-\alpha)n}{\alpha k}\binom{\alpha n}{(1-\alpha)k}} \right) - 2\log k - \log\log(en/k) \\
&\ge \log \left( \left( \frac{1-\alpha}{\alpha} \right)^{(1-2\alpha)k} \right) - 2\log k - \log\log(en/k) \\
&\ge (1-2\alpha)k \cdot \log \left( \frac{1-\alpha}{\alpha} \right) - 2\log k - \log\log(en/k).
\end{aligned}
$$

Therefore, we get our desired lower bound. $\qquad\square$

To sanity check, we can consider various choices of $\alpha$. For example, if we set $\alpha = 1/2$, we note that the space lower bound becomes trivially $0$. In fact, this makes sense as there are simple algorithms to obtain $\alpha = 1/2$ that require essentially no space. For example, we can use any random hash function $h$ that outputs random bits and return positive only when $h(x) = 0$. This obtains $\alpha = 1/2$ and essentially ignores the input set. Therefore, we can see that our lower bound is sensible.

Finally, we can use the above lemma combined with Theorem 4.1 to obtain our space lower bound for differentially private mechanisms that already require error probability.

*Proof of Theorem 4.2.* First, we apply Theorem 4.1 to get that the error probability $\alpha$ must satisfy

$$\alpha \ge \frac{1 - \delta}{e^\epsilon + 1}.$$

Note, for all choices $\epsilon \ge 0$ and $\delta \ge 0$, we see that $0 < \alpha \le 1/2$. Plugging in the error probability $\alpha \ge (1 - \delta)/(e^\epsilon + 1)$ into Lemma D.1, we get the following

$$\mathbf{E}[s] \ge \frac{e^\epsilon - 1 + 2\delta}{e^\epsilon + 1} \cdot k \cdot \log \left( \frac{1}{\alpha} - 1 \right) - O(\log k + \log\log n).$$

First, we note that $(e^\epsilon - 1)/(e^\epsilon + 1) = \Theta(1)$ for all choices of $\epsilon \ge 0$. For sufficiently large $k = \Omega(\log\log n)$, we get that

$$\mathbf{E}[s] = \Omega \left( \left( 1 + \frac{\delta}{e^\epsilon} \right) \cdot k \cdot \log(1/\alpha) \right)$$

completing the proof. $\qquad\square$

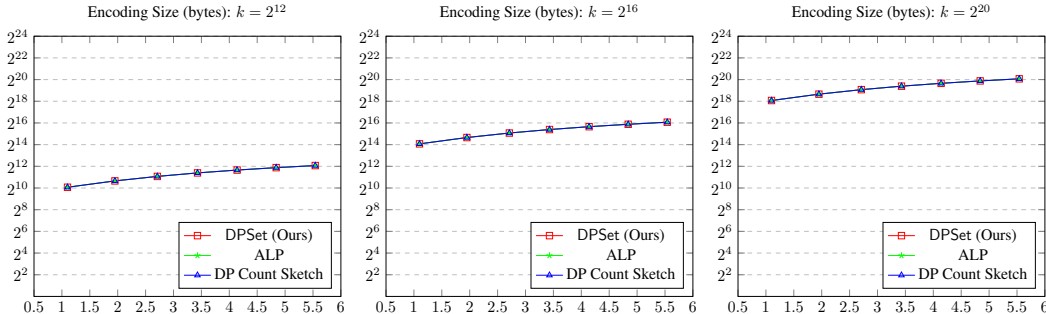

Figure 2: Comparisons of of DPSet, ALP, and DP Count Sketch with $\delta \leq 2^{-40}$. The $x$-axis is privacy parameter $\epsilon$ and the $y$-axis is encoding size in bytes.

# E  Space optimization

In our construction, we set the encoding size $m = (1 + \beta)k$ in Section 3. This is a worst-case guarantee to ensure that Solve may always be executed with the condition that the number of columns $m$ satisfies $m \geq (1 + \beta)n$ where $n$ is the number of rows (i.e., sampled set size $S'$). Instead, we can pick $m$ closer to the expected size of $S'$ and fail if it goes over.

For example, we can apply known probability tail bounds (such as Chernoff bounds) and pick $m$ to be closer to $(1 - p)k$ that is the expected size of $S'$. This increases the failure probability of Solve (and $\delta$) by an additive $e^{-O(k)}$ to account for if the number of rows is too large. Let $0 < \gamma \leq 1$ be a fixed constant. Invoking Chernoff bound, we see that the probability that $|S'| \geq (1 + \gamma)(1 - p)k$ is bounded above by $e^{-O(k)}$. Suppose that we assume that $|S'| \leq k' = (1 + \gamma)\frac{1-2\alpha}{1-\alpha}|S|$ and choose $m = (1 + \beta)k'$. This increases the failure probability of Solve by an additive $e^{-O(k)}$, which is very small. As this optimization increases $\delta$, it cannot be used for pure differentially private schemes.

Unfortunately, this ends up being a theoretical improvement as we were unable to empirically observe space efficiency gains in natural settings.

# F  Trivial algorithm for large error Probability

If we consider the case of large error probabilities $\alpha \geq 1/2$, there are trivial algorithms for differentially private subsets that use, essentially, no space and has perfect privacy guarantees of $\epsilon = \delta = 0$. In fact, it suffices to simply consider the case with error probability $\alpha = 1/2$.

Consider the following construction that completely ignores the input subset $S$. Pick a random hash function $h : U \to \{0, 1\}$. We represent the input subset $S$ using $h$. For any element $u \in U$, the decoding algorithm returns $\mathbf{1}_{h(u)=1}$. In other words, the decoding algorithms returns a uniformly random bit for each element $u \in U$. It is not hard to see that the error probability of this construction is exactly $1/2$. As the hash function $h$ is chosen independent of the input subset $S$, it is quite clear that this trivial algorithm achieves perfect privacy of $\epsilon = 0$ and $\delta = 0$. Therefore, all the constructions in our work focus on the case when $\alpha \leq 1/2$.

**Theorem F.1.** *There exists a perfectly secure $(0, 0)$-DP set mechanism with error probability $\alpha = 1/2$ where the encoding consists of a single hash function independent of the input set size.*

# G  Experimental evaluation of encoding size

We present graphs in Figure 2 showing the encoding sizes used in our experimental evaluation in Section 5. Recall that, for the purposes of comparing utility, we chose parameters such that all three constructions have similar encoding sizes. Therefore, the encoding sizes are essentially the same for all three constructions: DPSet from our work, ALP from [2] and DP Count Sketch from [34].

