# OpenReview forum: "Differentially Private Set Representations"
_NeurIPS.cc/2024/Conference — NeurIPS 2024 poster_

### Official Review · Reviewer_ys34 · 2024-07-04

**Soundness:** 3
**Presentation:** 3
**Contribution:** 4
**Rating:** 7
**Confidence:** 4

**Summary:**

The paper considers private representations of sets (under the neighboring relation of adding/removing an element). The goal is to answer set membership queries correctly with some nontrivial 1-alpha probability of being correct. (For a universe of size 1 this can be addressed by randomized response, so this problem generalizes releasing a single private bit.) Of course, the privacy guarantee will need to depend on alpha. Previous related work considered histograms (or multisets) and private versions of Bloom filters, but the performance of these past approaches on set membership queries has not been studied. The paper takes an interesting new angle, using ideas from space-efficient filters to get differentially private set representations using small space. It also gives lower bounds that nearly match the upper bound. Experimental comparisons to existing techniques suggest a better privacy-utility trade-off with comparable time usage.

**Strengths:**

Strengths:
- Studies an interesting special case of a well-studied problem (private histograms) and gives better guarantees for this special case
- An interesting generalization of randomized response to a "sparse" setting
- Literature review is comprehensive and well-written
- The techniques are interesting and novel

**Weaknesses:**

Weaknesses:
- It took me a while to understand why the results hold, the writing could be better
- It seems that the lower bound in Theorem 1.4 cannot possibly hold as stated (see questions below)

**Questions:**

- Can you make the privacy proof clearer? I think there are subtleties that should be discussed in the *main part* of the paper. For example, it would be good with a discussion of the probability of returning (⊥,⊥,⊥,S) when the linear system does not have full rank. This seems like a blatant privacy violation, though it happens only with probability 𝛿, so does not contradict approximate DP.
- Are there not some lower order terms missing in the space usage stated in Theorem 1.1 and 1.2? It seems that if you need to choose the field size a bit larger than 1/alpha, more bits per entry will be needed. Also, if the field size is at least e^epsilon it seems that each entry requires ceiling(epsilon/ln(2)) bits to encode.
- If you do not sample (i.e., use p=1), could you apply randomized response to the vector b before solving the linear system to achieve DP?
- If you return (⊥,⊥,⊥,S) with probability 𝛿 even when the linear system is solvable, does it yield pure DP?
- In some places (e.g. line 470) it is stated that the field size is 1/alpha. I believe you simply need it to be *at least* 1/alpha, as stated in line 261?
- Theorem 1.4 seems to be missing a condition that alpha is not too close to zero (or that epsilon is not too large)? The proof technique in the appendix does not seem able to show any space bound larger than the entropy of a random set of k elements from a universe of size n.
- Can you confirm the limitation mentioned below?

**Limitations:**

It should be made clearer that the DP guarantee relies on the hash functions being fully random while the space guarantees relies on the hash functions being pseudorandom. It would perhaps be more correct to state the results as computational DP.

---

> ### Author Rebuttal · Authors · 2024-08-06
>
> We thank the reviewer ys34 for their valuable and comprehensive feedback.
>
> In response to the reviewer's suggestion, we will improve the clarity of the privacy proof and incorporate additional context within the main body of the paper. We acknowledge the importance of explicitly explaining why returning $(\perp, \perp, \perp, S)$ is acceptable when the linear system cannot be solved, and why this does not violate the definition of approximate differential privacy. It is worth highlighting that returning $(\perp, \perp, \perp, S)$ is only a matter of convenience to simplify the privacy proof. In practice, alternative measures can be taken to prevent the release of the set in plaintext. For example, the algorithm could retry with newly sampled $h$ and $\mathsf{Row}$ until solving the linear system succeeds.
>
> The reviewer is correct with our Theorem 1.1 and Theorem 1.2 missing lower order terms. The space usage of the approximate DP scheme should instead be roughly ~$1.05 \cdot k \cdot \epsilon \cdot \lg(e)$ bits, and roughly ~$k \cdot \epsilon \cdot \lg(e)$ bits for the pure DP schemes. We thank the reviewer for pointing this out, and we promise to fix it in the next iteration of our paper.
>
> The reviewer also raised an interesting question on the possibility of using the randomized response on the RHS vector instead. In fact, we have explored this direction as well, but we encountered some challenges with either the query correctness probability and/or with the privacy proof. However, we agree that this is an interesting modification to the algorithm and worth exploring further in the future.
>
> The reviewer is correct that the field size just needs to be at least $1 / \alpha$ instead. We will make this clearer in the next iteration of our paper.
>
> We appreciate the reviewer's observation that Theorem 1.4 may not hold true when $\alpha$ is sufficiently small. Our current lower bound states that, in such cases, we would require more than $\log \binom{n}{k}$ bits, which is evidently incorrect - we can clearly represent any $k$-subset with 0 error probability with $\log \binom{n}{k}$ bits. To address this, we will modify the theorem statements to instead require at least $\min(\Omega((1 + \delta / e^\epsilon) \cdot k \cdot \log(1 / \alpha)), \log \binom{n}{k})$ bits. This correction will be incorporated in the next iteration of our paper. It is worth noting that our lower bound proof explicitly makes this assumption on Line 537 (but does not state in the theorem statement), and so our lower bound result is not invalidated.
>
> Finally, the reviewer is correct with the limitation of our work. Indeed, our DP guarantee and the space usage do rely on the hash functions being fully random and/or pseudorandom. If we assume pseudorandom hash functions (PRFs), we indeed obtain computational DP.

---

> > ### Comment · Reviewer_ys34 · 2024-08-08
> > **Thank you**
> >
> > I don't have further questions

---

### Official Review · Reviewer_VHxw · 2024-07-06

**Soundness:** 2
**Presentation:** 3
**Contribution:** 3
**Rating:** 5
**Confidence:** 3

**Summary:**

The paper studies the problem of optimal differentially private representing a set $S$ with size $k$ on universe $U$, under the setting where the set size $k$ is significantly smaller than the universe size $|U|$. In such settings, the authors propose algorithms to compute $(\varepsilon, \delta)$-DP set representation with error probability $\frac{1}{e^\varepsilon + 1}$, while only consuming $O(k\varepsilon)$ space and $O(k\log(1/\delta))$ encoding/decoding time. The algorithm is based on representing the set via solutions to linear equation systems formed by computing random hash functions on each set member. The authors further prove that under $(\varepsilon, \delta)$-DP, their upper bound matches lower bounds in both error probability and space requirement, up to a factor of $\log(1/\delta)$.

**Strengths:**

- The paper has a clear presentation, and the results are relatively complete in matching lower and upper bounds.
- The approach of encoding set membership via solution to linear equation systems is interesting, and its application to differentially private set representation is novel to the best of my knowledge

**Weaknesses:**

- I find that one part of the proof for error probability needs more clarification. Specifically, in lines 465-466, the authors write "the false negative probability is s not sampled into S′ and the linear constraint being unsatisfied that is $p(1-F^{-1})$". However, to my understanding, this probability should be $(1-p)(1-F^{-1})$. Maybe the authors could clarify more whether this is a typo and whether it affects any results.

- The proof of error probability $\alpha$ requires the existence of a $1/\alpha$-finite field, which limits the applicability of the proposed algorithm for small $\varepsilon$. For example, the algorithm's error upper bound appears to be identical for all $\varepsilon \leq \ln(2)$, as under such settings the smallest finite field that is larger than $1 + e^{\varepsilon}$ is always $Z_3$.

- Executing the algorithm requires knowledge of an upper bound of the set size $k$, as the algorithm requires to form an underdetermined linear equation system. This may significantly restrict the applicability and optimality of the proposed algorithm when no such knowledge is available.

**Questions:**

Besides the points listed in the weakness, I have the following questions related to novelty and presentation.

- Is the non-DP variant of the set representation algorithm in the paper proposed in prior literature, or is it a novel algorithm?  This is to understand the novelty of the proposed algorithm.
- In Theorem 1.1 and 1.2, the pure DP encoding time is $O(k(\log k)^2)$ while the $(\varepsilon, \delta)$-DP encoding time is $O(k\log(1/\delta))$. Considering that typically we have $\delta\ll 1/k$, this suggests that the pure DP algorithm needs smaller encoding time, which is a bit counterintuitive as pure DP is a stronger requirement. Could the authors clarify why this happens?
- The decoding Algorithm 2 appears to require new computation for each element of the universe $U$. However, in Theorem 1.1 and 1.2, the decoding time only grows with set size $k$ and does not depend on the universe seize $|U|$. Could the authors clarify this discrepancy?


Minor typo:
- line 221, $n \geq k$ should be $m \geq k$, "full rank" should be "full column rank"

**Limitations:**

Yes

---

> ### Author Rebuttal · Authors · 2024-08-06
>
> We thank the reviewer VHxw for their valuable and constructive feedback.
>
> We will first address the reviewer's question regarding the error probability. We think that the proof presented in our paper is accurate. We believe the misunderstanding may arise from the way the pseudocode in Algorithm 1 is structured. In Algorithm 1, we insert each element of set $S$ into set $S'$ with a probability of $1 - p$. Note that this is equivalent to removing each element from $S$ with a probability of $p$ and considering the resulting set as $S'$. Consequently, the probability of a false negative is indeed $p(1 - 1 / |F|)$, because a false negative can only occur if the element is not present in $S'$ (e.g. removed) and the false positive does not occur for this removed element. We will try to rewrite the pseudocode to make this clearer.
>
> The reviewer raised a question if the non-DP set representation algorithm in our paper had been discussed in earlier research. To clarify, some of the linear systems in our paper have been used to construct non-DP sets while others have not. Random band matrices have been studied as non-DP sets (we mention this in the first paragraph of Section 3). On the other hand, Vandermonde matrices have never been used to build non-DP sets to our knowledge. However, we want to emphasize that the novelty of our work lies in our observation that any linear system with certain properties can be modified to produce differentially private set representations. Our work further proposes a general DP framework based on these linear systems. We believe this observation is highly significant and non-trivial.
>
> We note that there are also non-DP set data structures (like cuckoo filters) that are not directly compatible with our framework, indicating that our observation is non-trivial. To our knowledge, we are unaware of any method to convert them into DP sets. In particular, there are certain properties unique to the linear system set data structures that we rely on to ensure DP privacy guarantees.
>
> The reviewer also requested clarification regarding the encoding runtimes of the pure DP and the approximate DP algorithms. Indeed, the results may appear counterintuitive. The distinction between the two algorithms lies in how they solve linear systems.
>
> For the pure DP algorithm, the Vandermonde matrix's structure allows for the use of an FFT-like algorithm to solve the linear system efficiently. However, this approach is not feasible for the random band linear systems in the approximate DP algorithm, and we thus use the Gaussian elimination instead. While Gaussian elimination typically takes $O(n^3)$ on a general linear system, we emphasize again that the random band linear system’s unique structure allows the linear system to be solved very efficiently n $O(n  \log(1/\delta))$, both theoretically i and practically.
>
> It is worth noting that this difference in matrix structure also impacts the decoding times. The pure DP decoding algorithm has a time complexity of $O(k)$, while the approximate DP decoding algorithm has a time complexity of $O(\log(1 / \delta))$. In this respect, the pure DP construction offers pure differential privacy and efficient encoding at the cost of a slower decoding time.
>
> In relation to the reviewer's third question, we emphasize that the decoding time of $O(k)$ and $O(\log(1 / \delta))$ applies to decoding/querying a single element. Consequently, if we query $T \subseteq U$ elements, the total decoding time becomes $O(|T| \cdot k)$ and $O(|T| \cdot log(1 / \delta))$, respectively. Our formulation of the decoding time is consistent with previous studies on the differentially private $k$-sparse vector problem (e.g. Table 1 in [1]). In these works, the decoding time is expressed in terms of the time required to access a single entry of the $k$-sparse vector. However, we acknowledge that the term “decoding” can be ambiguous, and so we will change it to “access” or “query” instead in the next iteration.
>
> [1]: https://arxiv.org/pdf/2106.10068

---

> > ### Comment · Reviewer_VHxw · 2024-08-13
> >
> > Thanks for the reply, which partially addresses my concerns, except for weaknesses 2 and 3. Thus I will keep my rating as is currently.

---

### Official Review · Reviewer_1hbz · 2024-07-11

**Soundness:** 1
**Presentation:** 3
**Contribution:** 3
**Rating:** 5
**Confidence:** 4

**Summary:**

The paper addresses the problem of releasing a set $S $ of $ k $ elements from a potentially very large universe $ U $ in a differentially private manner. Here, two input sets $ S $ and $ S' $ are considered neighboring if their symmetric set difference is at most one; that is, $ S $ and $ S' $ differ by adding or removing exactly one element.

The objective is to publish a concise representation of the elements in $ S $ that allows determining whether an element in $ U $ belongs to $ S $, while minimizing both false positives and false negatives. The paper introduces new algorithms for constructing succinct representations using solutions from random linear systems based on the elements in $ S $.

The $(\epsilon, \delta)$-differentially private construction achieves an error probability of $\frac{1}{e^\epsilon + 1}$, uses space of at most $1.05k\epsilon$ bits, has an encoding time of $O(k \log(1/\delta))$, and a query time of $O(\log(1/\delta))$ per element.

On the other hand, the $\epsilon$-differentially private construction maintains the same error probability, uses space of at most $k\epsilon$ bits, but requires $O(k)$ query time per element. The space usage of both constructions matches the proposed lower bound up to constant factors.

**Strengths:**

1. The constructions appear novel. Rather than directly randomizing set $ S $, the paper utilizes elements from $ S $ to establish random linear constraints and publishes a solution that satisfies these constraints (demonstrating its existence).

2. The space usage aligns with the lower bound.

3. The algorithms demonstrate efficiency: both approaches exhibit slightly more than linear encoding time, and the approximate differentially private algorithm achieves $ \tilde{O}(1) $ query time.

**Weaknesses:**

It seems there's a concern regarding the privacy analysis in the paper, specifically related to the use of the set $ S $ of size $ k $ to generate random linear constraints. Let's clarify and address the issues raised:

1. **Number of Linear Constraints $ m $**: It's mentioned that $ m = (1 + \beta) k $ for some constant $\beta$. This implies that the number of constraints $m$ depends on $k$.

2. **Dimension of Published Solution $ x $**: The solution $ x $ that is published has a dimension of $ m $, which is dependent on $ k $ as discussed.

3. **Privacy Concern**: Given the definition of neighboring datasets (differing by adding or removing one element from $ S $), an adversary could potentially distinguish between these datasets by observing the dimension of the output vector $ x $. This suggests a potential privacy vulnerability if the dimension of $ x $ reveals information about the dataset $ S $.

To address this issue, possible fixes might impact current the error probability and subsequently affect the space usage analysis.

**Questions:**

Can the embeddings proposed in this paper be mergable? For example, if we have two representations $ x_S $ and $ x_T $ created by the algorithms in this paper for sets $ S $ and $ T $, can we directly create a representation $ x_{S \cup T} $ for $ S \cup T $ using $ x_S $ and $ x_T $?

**Limitations:**

See weakness.

---

> ### Author Rebuttal · Authors · 2024-08-06
>
> We thank reviewer 1hbz for their constructive feedback.
>
> The reviewer raised a concern about the privacy analysis of our paper, suggesting that the size of the published vector could compromise privacy.
>
> We want to clarify that the parameter $k$ is an upper bound on the size of the set $S$ to be encoded. In other words, we treat $k$ as an algorithm parameter, and the algorithm only accepts an input set $S$ with $|S| \leq k$. Therefore, the dimension of the published vector is fixed to $m = (1 + \beta)k$ for all valid input sets $S$ of size at most $k$, and the privacy proof holds in this scenario.
>
> We note that this aligns with prior works ([1], [2]) that studied differentially private releases of $k$-sparse vectors. Here, a $k$-sparse vector is defined as a vector with at most $k$ non-zero entries (see first paragraph of Section 2 in [1] and second paragraph of page 2 in [2] for the definition of $k$-sparse vectors). Furthermore, the notion of DP is only defined over $k$-sparse vectors (see Definition 2.1 in [1] for example). Our work can be seen as studying a more specific instance of a differentially private $k$-sparse vector problem where the non-zero entries are restricted to values of one. In prior works, the space usages of the algorithms were stated in terms of $k$ (an upper bound on the number of non-zero entries), which aligns with how we state the space usages of the algorithms in our work (see Table 1 in [1] and [2]).
>
> Upon reviewing our paper more carefully, we acknowledge that this point was not clearly stated. We commit to addressing this in the next iteration of our paper to ensure clarity.
>
> The reviewer also posed an intriguing question: "Can the two published vectors $x_S$ and $x_T$ (encoding $S$ and $T$, respectively) be merged as $x_{S \cup T}$ to encode the union set $S \cup T$?" In fact, we have considered this exact problem, but unfortunately, we could not find a suitable solution. For now, we leave this as an interesting open research question.
>
> [1]: https://arxiv.org/pdf/2106.10068
> [2]: https://arxiv.org/pdf/2112.03449

---

> > ### Comment · Reviewer_1hbz · 2024-08-13
> >
> > Thank you for your response.
> >
> > We recommend updating the definitions accordingly and revisiting the paper to modify the descriptions of $S$ wherever it appears. For instance:
> >
> > 1. In the abstract, line 2: "...sets of size $k$ from..."
> > 2. Line 235: "Suppose we are given an input set $S = \{s_1, \dots, s_k\} \subseteq U$ of size $S| = k$."

---

### Official Review · Reviewer_Gmoe · 2024-07-12

**Soundness:** 3
**Presentation:** 4
**Contribution:** 2
**Rating:** 6
**Confidence:** 3

**Summary:**

The paper presents new differentially private (DP) mechanisms for representing sets of size k  from a large universe. It introduces two algorithms: one for epsilon, delta -DP representations and the other for pure epsilo-DP representations, with faster decoding. Both algorithms achieve optimal privacy-utility trade-offs and match new space lower bounds up to small constant.

**Strengths:**

* The work is novel and appears to be correct, though I did not read and check the proofs for the theorems in great detail.
* Theorem statements are clear, and are presented in a transparent manner.
* Experimental results are presented clearly and align with the theoretical work.

**Weaknesses:**

* I think that the technical work is strong, but that the use case of this kind of work remains a bit unclear to me. If the authors could spend more time motivating the need for DP set representations and specific use cases, this would help contextualize the work better. The authors cite that this is useful for applications where users wish to privately disclosure sets of bookmarked websites - can this example be fleshed out more. What would the privacy attacks, and who would the adversary be? This helps structure the work in the context of local vs central DP, etc.
* This also connects with the related works, as I find it somewhat difficult to understand the exact use case of DP set representations, it was hard for me to understand what gaps the related works didn’t fill. Even if the related works sections are well-written.
* Some work in section 3 could be introduced a bit more, for example the work on “Random Band” and “Vandermonde” matrices. I don’t think it’s clear (at least to me)why this design choice was made in the pure versus approximate DP setting. This would help with understanding the gap between the the pure and approximate DP setting as well.

**Questions:**

* In Algorithm 1, could you explore the failure modes of the algorithm more?
 * Theorem 3.1 - F is introduced, but it is not directly introduced as an input to algorithm 2. There’s some organizational work on that would help with comprehension.

**Limitations:**

Yes.

---

> ### Author Rebuttal · Authors · 2024-08-06
>
> We thank the reviewer Gmoe for their valuable feedback.
>
> To illustrate the usefulness of our schemes, we will revisit the installed apps use case briefly mentioned in the paper and provide a concrete example. Imagine an analyst at an app software company looking to gather statistics on the percentage of a specific population who have installed apps developed by their company in a differentially private manner. One way to achieve this would be to use the randomized response scheme, which could take the form of the following question on the set of candidate apps: "Does this device have app Y installed?". Then, using the probability parameter $p$, the analyst can infer how many devices have installed these candidate apps.
>
> However, now imagine that the set of candidate apps is very large, say on the order of thousands (perhaps, the analyst also needs to get statistics on competitors’ apps) . This is much more than the number of apps installed on a typical device. In this case, asking and retrieving answers to the questions may incur significant bandwidth overhead. One could use non-DP sets, but that would directly reveal all installed apps to the analyst.
>
> One solution to this problem is to employ our scheme. We configure an upper bound $k$ on the number of apps installed on the device (say, 250), then run our encoding algorithm to release the differentially private representations of the apps from the devices (suppose that we ignore devices with more than 250 apps installed, which are considered outliers). The analyst can invoke the decoding algorithm on the candidate apps to determine the distribution of the installed/uninstalled, and derive accurate statistics on the percentage of devices with these apps installed. While the differentially private app representation includes more information compared to a basic randomized response approach (as it encompasses data on other apps), it can offer a practical balance between privacy and utility in certain scenarios. Note that our scheme still ensures that the analyst cannot exactly identify the set of installed apps on each device.
>
> In response to the reviewer’s suggestion, we will elaborate more on the constructions of “Random band” and “Vandermonde” constructions in the next iteration of our paper. The primary distinction between our pure DP and approximate DP schemes lies in the failure probability associated with the constructed linear systems. When utilizing the Vandermonde matrix construction, the constructed linear system is always solvable, resulting in a pure DP scheme (it never outputs $(\perp, \perp, \perp, S)$). In contrast, using the random band matrix construction introduces a small probability of failing to solve the linear system, leading to a non-zero probability of outputting $(\perp, \perp, \perp, S)$.
>
> While outputting $(\perp, \perp, \perp, S)$ in this scenario may appear to be a blatant privacy violation, it is important to note that the probability of this occurring is small, as it is bounded by $\delta$ in the approximate DP definition. Therefore, it satisfies the approximate DP requirements.
>
> Furthermore, it is crucial to emphasize that outputting $(\perp, \perp, \perp, S)$ solely serves as a convenience in the privacy proof. In practice, alternative measures can be taken to ensure that the set is not disclosed in plaintext. For example, the algorithm can simply retry with new $h$ and $\mathsf{Row}$ until the constructed linear system is solvable.
>
> Finally, we thank the reviewer for suggesting improvements to the presentation of the paper. We promise to incorporate the feedback in the next iteration of our paper.

---

### Decision · Program_Chairs · 2024-09-25

**Decision:**

Accept (poster)

**Comment:**

The paper presents interesting novel technical work on a special case of the well-studied problem of releasing histograms under differential privacy. The reviewers had some concerns about the quality of the exposition in the paper, and the clarity of the motivation of the problem they study. The work also has some limitations, e.g., the need for the public upper bound $k$ on the sparsity of the histogram. Nevertheless, the paper is an interesting contribution.